# Tellurium and Nano-Tellurium: Medicine or Poison?

**DOI:** 10.3390/nano14080670

**Published:** 2024-04-12

**Authors:** Daniella Sári, Aya Ferroudj, Dávid Semsey, Hassan El-Ramady, Eric C. Brevik, József Prokisch

**Affiliations:** 1Nano-Food Laboratory, Department of Animal Husbandry, Institute of Animal Science, Biotechnology and Nature Conservation, Faculty of Agricultural and Food Sciences and Environmental Management, University of Debrecen, 138 Böszörményi Street, 4032 Debrecen, Hungary; ferroudj.aya@agr.unideb.hu (A.F.); semi@gmail.hu (D.S.); jprokisch@agr.unideb.hu (J.P.); 2Soil and Water Department, Faculty of Agriculture, Kafrelsheikh University, Kafr El-Sheikh 33516, Egypt; 3College of Agricultural, Life, and Physical Sciences, Southern Illinois University, Carbondale, IL 62901, USA; eric.brevik@siu.edu

**Keywords:** nano-medicine, nano-toxicity, medicinal attributes, therapeutics, antimicrobial agents

## Abstract

Tellurium (Te) is the heaviest stable chalcogen and is a rare element in Earth’s crust (one to five ppb). It was discovered in gold ore from mines in Kleinschlatten near the present-day city of Zlatna, Romania. Industrial and other applications of Te focus on its inorganic forms. Tellurium can be toxic to animals and humans at low doses. Chronic tellurium poisoning endangers the kidney, liver, and nervous system. However, Te can be effective against bacteria and is able to destroy cancer cells. Tellurium can also be used to develop redox modulators and enzyme inhibitors. Soluble salts that contain Te had a role as therapeutic and antimicrobial agents before the advent of antibiotics. The pharmaceutical use of Te is not widespread due to the narrow margin between beneficial and toxic doses, but there are differences between the measure of toxicity based on the Te form. Nano-tellurium (Te-NPs) has several applications: it can act as an adsorptive agent to remove pollutants, and it can be used in antibacterial coating, photo-catalysis for the degradation of dyes, and conductive electronic materials. Nano-sized Te particles are the most promising and can be produced in both chemical and biological ways. Safety assessments are essential to determine the potential risks and benefits of using Te compounds in various applications. Future challenges and directions in developing nano-materials, nano-alloys, and nano-structures based on Te are still open to debate.

## 1. Introduction

Tellurium (Te) is important due to its involvement in certain alloys, producing photovoltaic solar cells [1], thermoelectric cooling devices [2], ceramics and glass [3], and machines, printers, and telescopes [4]. Tellurium is characterized as a p-type semi-conductor with a narrow band gap energy of 0.34 eV. Due to its anisotropic crystalline structure, Te displays a multitude of distinct properties, including photoconduction and piezoelectricity, and can be used in thermoelectric, metallurgy, and manufacturing applications [5,6]. Tellurium is located in group 16 of the periodic table, which is known to hold both the chalcogen and oxygen groups. The atomic number of tellurium is 52. Tellurium is situated below selenium and above polonium in the periodic table. Tellurium is a metalloid that is brittle, silver-white, and behaves as a semiconductor [7]. Tellurium exists in four oxidation states (−2, 0, +4, and +6) and forms compounds that are chemically similar to those formed by sulfur and selenium. Tellurite (Te(IV)) is 100 to 1000 times more toxic than selenite, which is chemically similar. Additionally, the toxicity of Te(IV) is up to 10 times higher than that of Te(VI) [8].

Several isotopes of Te occur naturally. Those that are fairly common are ^122^Te, ^124^Te, ^125^Te, ^126^Te, ^128^Te, and ^130^Te at 2.5, 4.6, 7.0, 18.7, 31.8, and 34.5%, respectively. Tellurium, like sulfur and selenium, is able to adopt oxidation states ranging from −2 to +6, and poly-tellurium cations and anions both exhibit several fractional oxidation states [4]. Higher oxidation states are more stable for selenium and tellurium, in part as a result of lower ionization energies. Tellurium tends to catenate; however, this is less pronounced than in sulfur or selenium [9]. Tellurium appears in sulfide ore deposits with other chalcophile elements, such as silver, copper, and lead, rather than forming separate telluride minerals. Tellurium can substitute for sulfur in common sulfide minerals like chalcopyrite, galena, covellite, pyrrhotite, pyrite, and sphalerite [10]. Nano-tellurium has promising applications, including antimicrobial [11,12], electronic, and optoelectronic [13] uses. Nano-tellurium (nano-Te) can be found in many forms such as nano-Te structures [13], nano-Te wires [14], nano-Te tubes [15], and nano-Te ribbons [16]. Nano-Te can be made with chemical methods along with biological and green methods that are considered eco-friendly. Nano-Te has potential applications in agriculture, medicine, bioengineering, and bioremediation [17,18,19,20]. Common terms associated with nano-tellurium that are important to understand in this review are given in Table 1. 

While Te and nano-Te are important in modern industry, have been shown to influence human health, and can accumulate in the food chain, there are still many unanswered questions about Te and nano-Te. Therefore, this review focuses on Te and nano-Te from many points of view, including its discovery, occurrence, forms in nature, chemical interactions, and production. This review will also highlight potential applications and toxicity issues of Te and nano-Te relating to human health and discuss future needs.

## 2. Discovery of Tellurium

Tellurium was discovered in the 18th century. Three different investigators discovered it virtually simultaneously. The discovery of Te is connected to Hungary because it was identified in samples from Transylvania, which was part of Hungary at the time [21]. In 1782, a Swedish scientist named Franz-Joseph Müller von Reichenstein discovered what he believed was a new element while conducting experiments on a mineral sample from Transylvania that he named ‘aurum paradoxum’ due to its unusual properties (Figure 1A). However, further research revealed that this “element” was actually a compound of other elements and an unknown substance found in gold ore [21]. In 1798, Martin Heinrich Klaproth received a sample of the same mineral from Reichenstein and decided to investigate it further (Figure 1B). Klaproth analyzed the sample and managed to isolate a new element, which he named “tellurium” after the Latin word “tellus”, meaning “earth” [21]. Also in 1798, at almost the same time Klaproth was working on isolating tellurium, the Hungarian chemist and botanist Pál Kitaibel independently discovered the element in a gold ore sample from the Nagybörzsöny mine in Hungary (Figure 1C) [21].

## 3. Tellurium’s Occurrence, Forms in Nature, and Characterization

### 3.1. Tellurium’s Occurrence

Tellurium is very rare in Earth’s crust, but it has an impressive mineralogical diversity with over 160 known mineral species containing tellurium. Many of these minerals exhibit unique crystal structures [25]. Tellurium is more abundant in the universe than any other element with an atomic number greater than 40. However, when it comes to the Earth’s lithosphere, it is among the least abundant elements, and it is notably absent from seawater, with only a trace presence. In stark contrast, Te is the fourth most plentiful trace element found in the human body, following iron (Fe), zinc (Zn), and rubidium (Rb). Furthermore, it is unusually prevalent in the human diet. This peculiar situation, in which Te is abundant in human food but scarce in soil, suggests that plants have evolved highly efficient mechanisms for absorbing Te through their roots compared to other trace elements [26]. 

### 3.2. Global Tellurium Production

Currently, Te is primarily mined from two regions. One is in Southwestern China, specifically the Dashuigou and Majiagou gold–tellurium epithermal vein deposits. The other one is the Kankberg deposit in Västerbotten County (Sweden), which exhibits epithermal-like mineralization. Together, these deposits contribute approximately 15% (around 70 metric tons) of the annual worldwide production of Te, which ranges between 450 and 470 metric tons [27]. Globally, it is predicted that 9500 metric tons of Te have been deposited near Cu smelters since 1900 [28]. In a remote area of Canada, modern Te deposition rates were six times higher than preindustrial rates, indicating a significant increase in atmospheric Te levels due to human activities. There are several industrial plants in which this kind of recovery of Te is achieved, including the Vale’s CRED plant in Canada, the Luilu metallurgical plant in Congo, the Freeport refinery in the USA, Naoshima’s Smelter and Refinery in Japan, and numerous major copper smelters in China, including Tongling Nonferrous Metals Smelter and Zhongyuan Gold Smelter [29]. The main tellurium producers in the world include China, Russia, Japan, Canada, Sweden, and others, rendering about 640 tons in 2023 (Figure 2). The main global Te import sources from 2019 to 2022 were Canada (38%), Germany (34%), the Philippines (15%), Japan (6%), and others (7%) [30].

### 3.3. Tellurium Forms in Nature

Tellurium is found as free in nature but is most often found in certain ores such as calaverite (AuTe_2_), sylvanite (AgAuTe_4_), and krennerite (AuTe_2_). Tellurium occurs mainly as tellurides with gold, silver, lead, and bismuth, but presently, most produced Te comes as a byproduct from mining and refining copper [7]. Significant Te contamination has been found near copper (Cu) and Zn smelters, with an estimated 21 g of Te released per metric ton of Cu processed. Tellurium is mainly derived during the electrolytic refining of Cu [31]. During this refining, Te is removed by leaching solutions, which is typically accomplished through the reduction in Te(IV/VI) species using metallic Cu. This method results in the formation of insoluble copper telluride (Cu_2_Te). Tellurium has become an emerging environmental contaminant of concern, with its deposition influenced by its lake water residence time and biological processing [28]. Tellurium in soil is mainly associated with Fe(III) hydroxides under oxidizing conditions [32]. In ocean water, Te occurs predominantly as H_5_TeO_6_ [33]. Various analytical techniques have been employed to quantify tellurium in natural samples. In earlier studies, methods such as UV–visible spectrophotometry, neutron activation analyses, and graphite furnace atomic absorption spectrometry were commonly used. More recent research has leaned towards utilizing voltammetry, hydride generation atomic fluorescence spectrometry, and inductively coupled plasma mass spectrometry [34]. The types of Te compounds typically found in nature are organic Te-containing complexes and inorganic forms of Te. The common mineral forms of Te involve hydrogen telluride (H_2_Te), elemental Te (Te^0^), Te-monoxide (TeO), tellurite (TeO_3_^2−^), and tellurate (TeO_4_^2−^). Te complexes are compounds that have a central Te atom bound to a range of ligands like cysteine proteases. Organo-telluride compounds have behaviors that mimic the antioxidant glutathione peroxidase or vitamin E [35]. Other nano-forms of Te include tri-telluride quantum materials (e.g., cadmium telluride quantum dots).

### 3.4. Tellurium Characterization

Tellurium is classified as a metalloid and is solid at room temperature. The main physical and chemical properties of Te are presented in Table 2, Table 3 and Table 4. The crystalline form of Te has a silvery-white appearance, and when pure, it exhibits a metallic luster, is brittle, and can easily be pulverized. Amorphous Te can be obtained by precipitating Te from a solution of telluric [Te(OH)_6_] or tellurous acid (H_2_TeO_3_) [7]. Human exposure to Te is commonly characterized by a sour garlic odor in one’s breath, urine, and sweat. Garlic plants have the ability to absorb tellurate, an inorganic Te compound, and convert it into telluro-amino acid [36]. Among the general populace, Te exposure primarily arises from dietary sources, including meat, dairy items, and cereal products [37]. The presence of elemental Te nano-particles (Te-NPs) has been shown in regolith samples, shedding light on the biogeochemical behavior of Te in Earth’s surface environment. These Te-NPs were found in significant concentrations at locations near and far from Te ore weathering sites. Analytical techniques, including X-ray fluorescence mapping and X-ray absorption spectroscopy, have revealed that most Te in regolith samples is associated with clay minerals and Fe oxyhydroxides, primarily in the +IV and +VI oxidation states. While Te-NPs constitute a small fraction of total Te, their presence indicates the active biogeochemical cycling of Te in surface environments. Te-NPs are reactive and likely formed through microbially mediated reduction processes in distal samples [10]. 

## 4. Nano-Tellurium and Its Production

The importance of nano-technology is growing from year to year. Nano-technology connects physics, chemistry, biology, materials science, electronics, medicine, and agriculture. Engineered nano-particles (NPs) are receiving a growing amount of global attention due to their attractive properties, multifunctionalities, unique characteristics, and innovative applications in different industrial and scientific domains. There are several chemical, physical, and biological methods of producing NPs, such as laser ablation, coprecipitation, the hydrothermal route, the solvothermal route, the sol-gel process, the polyol process, sonochemistry, and microwave-assisted and electrochemical methods [17]. As an alternative approach, green chemistry focuses on creating materials from synthetic chemistry as an alternative to traditional synthesis. In green chemistry, living organisms like bacteria, human cells, fungi, and plants as well as dietary and natural organic compounds such as honey, tea, coffee extracts, or biological wastes produced from industrial food plants are used to naturally synthesize NPs [39]. The metallic NPs generated should be useful and have nano-metric structures that are effective in both industrial and health care applications. 

Vernet Crua and her co-workers compared two different synthesis methods for producing nano-Te wires (NWs), including a green method and a traditional chemical method [39]. The aim was to investigate the differences in the anticancer and cytocompatibility properties of Te-NPs generated through these two processes. They conducted experiments using both healthy fibroblasts (HDF cells) and cancerous melanoma cells, testing a range of NP concentrations from 5 to 100 μg mL^−1^. The results showed that when comparing green Te-NPs with chemical Te-NPs, there was an improvement in the proliferation of HDF cells and a decrease in the proliferation of cancerous cells. The green synthesis of Te-NPs demonstrated an enhancement in healthy fibroblast proliferation compared to that of chemical Te-NPs. This finding suggests the green synthetic approach offers advantages in terms of safety, cost-effectiveness, efficiency, and biocompatibility for a wide range of biomedical applications [39].

Cadmium-telluride (CdTe) quantum dots (QDs) were successfully synthesized using orange peel extract waste as an inorganic semiconductor material. The QDs were 6 nm [40]. Gómez-Gómez and co-workers found that the interaction between the nano-particles and bacterial communities caused the Te nano-particles to change their shape from spheres to rod-shaped particles. These findings provide new insights into the behavior of uncommon nano-particles, such as Te-based nano-particles, when they interact with microbial communities [41]. 

Tellurium is not essential in biological metabolism; moreover, it is toxic at very low concentrations. A list of studies on the production of nano-Te using biological approaches is given in Table 5. The K_2_TeO_3_ and Na_2_TeO_3_ precursors are less toxic than other forms of Te, so they can be used to produce Te-NPs. Te(^0^) can be produced naturally depending on the respiration of microorganisms (e.g., *Saccharomyces cerevisiae*); fermentation can increase this production. When oxygen is limited, NPs can be produced by bacteria. In Te-enriched *Spirulina platensis* cultures, Te-NPs were present. Te interacts with two phycobiliproteins: allophycocyanin and phycocyanin [17].

Bacteria reduce tellurium oxyanions, both in planktonic cells and biofilms. Microorganisms play a crucial role in altering the behavior of metals and metalloids in the environment. Notably, tellurite can be precipitated either inside or outside bacterial cells, serving as an electron sink to facilitate bacterial growth [50]. This discovery offers fresh insights for microbial physiologists and biotechnologists alike [50]. Abo Elsoud and co-workers [47] found six fungal isolates that are able to reduce K_2_TeO_3_ into elemental Te-NPs. *Aspergillus welwitschiae* proved to be the most effective fungal isolate connection for the biogenic production of Te-NPs (60.8 nm) with oval to spherical shapes. The produced Te-NPs were evaluated for antimicrobial and antibacterial activity at 25 mg ml^−1^ against *E. coli* and *Staphylococcus aureus* (MRSA) [47]. Tellurium is toxic to plants at high enough concentrations, but some plants have the ability to metabolize Te and transform it into telluro-amino acids and organo-tellurium. Garlic can produce Te-methyltellurocysteine and S-methyltellurosulfide metabolites. Most of these metabolites had a Te-NP size around 40–55 nm and were highly concentrated in the initial part of the roots and at the tips. Plant-produced Te-NPs tend to be sphere-, plate-, and rod-shaped [17].

## 5. Applications of Tellurium and Nano-Tellurium

Studies on the applications of Te and nano-Te tend to focus on photoelectricity, including photodetectors and field-effect transistors [13,38]. These applications rely on the fact that Te is considered the “vitamin of modern industry” in the 21st century due to its unique and intriguing properties [51]. The main applications of Te are depicted in Figure 3. These applications involve the biomedical, biological, and industrial fields due to Te-NPs’ optical, electrical, and thermoelectric properties with an outstanding level of environmental stability [52]. There are several new industrial applications for nano-Te compounds like cadmium telluride (CdTe) in solar panels, thermoelectric devices, batteries, and nano-materials such as CdTe QDs [53]. Renewable energy sources are highlighted as a way to address anthropogenic climate change, making the use of CdTe in solar panels very advantageous. Cadmium telluride solar panels currently supply 5% of the global solar panel market [54].

Important properties of nano-Te structures include their controllable band gap, outstanding electrical properties, and good environmental stability. These nano-structures can be formed in zero- (0D), one- (1D), two- (2D), three- (3D), and complex dimensional nano-forms (Table 6) [13]. Many Te nano-structures can be synthesized to have various controllable shapes, sizes, and structures and are considered to be next-generation in optoelectronic devices and electronics (Figure 4). Tellurium NPs have unique low-dimensional chemical and physical properties with special structures and small sizes compared with those of traditional 3D NMs. Low-dimensional NMs have great development potential in the fields of photoelectric conversion, nonlinear optics, electronics, magnetic transport, and biomedicine [13]. Generally, zero-dimensional NMs refer to nano-structures with a size less than 100 nm, whereas 1D NMs have unique physical, structural, and chemical properties, such as nano-tubes, nano-wires, and nano-ribbons with one dimension that may exceed 100 nm [55,56,57,58].

The global photovoltaic (PV) industry is experiencing rapid growth due to the decreasing cost of PV modules, growing concerns about the limited availability of fossil fuels, and increasing awareness of greenhouse gas emissions and their environmental impact [59]. The industry is also marked by innovations in various types of PV technologies. Crystalline silicon has historically dominated the PV panel production market. However, thin-film PV technologies, such as copper indium gallium di-selenide (CIGS), cadmium telluride (CdTe), and amorphous silicon, have gained traction due to improvements in energy conversion efficiency and cost reductions. The energy conversion efficiency of Te alloys used in the PV industry depends on the element that is combined with Te (Table 7). The annual power production from CdTe panels has grown significantly in recent years, outpacing the growth of other thin-film technologies and making CdTe a focal point in the analysis of thin-film PV panels [60]. Tellurium has various other industrial applications, such as serving as a catalyst and pigment in ceramics and an additive to other metals in metallurgy, as well as being used in glass optical fibers for telecommunications and playing a role in the manufacturing of magnetic disks [61]. Tellurium alloys are used in nano-materials, including QDs [62]. In rubber production, Te is used in the vulcanization process to enhance resistance to heat, abrasion, and aging. In metallurgy, Te serves as an additive in the production of cast iron, steel, and copper [63]. It helps improve machinability and enhances the surface’s resistance to wear and corrosion. In glass manufacturing, Te is utilized as a coloring agent [3], while it is used as a catalyst in the chemical industry. Tellurium is also used in the tellurite diagnostic test for diphtheria and has therapeutic applications as well.

Bulk-Te compounds are relatively limited in their applications due to Te’s high toxicity, whereas nano-Te forms have shown promising applications in the pharmaceutical and biomedical fields (Figure 4). Their biomedical applications may include their antimicrobial [12], antifungal [74], antibacterial [75], and anticancer [76] activities, their use as therapeutic agents [77], and their use in imaging applications [17,38,78]. Therefore, Te-NMs are presented as a new field that can be used in the biomedical and therapeutic industry as a possible alternative to other well-known metallic NPs such as silver and gold [38].

### 5.1. Pharmaceutical Applications

Potassium tellurite (K_2_TeO_3_) has been used to treat various medical conditions including syphilis, cystitis, eye infections, tuberculosis, dermatitis, and leprosy. Te-containing soluble salts had a historical role as therapeutic and antimicrobial agents before the advent of antibiotics [79]. Despite the fact that organo-Te compounds have been synthesized for over 150 years, our understanding of their pharmacological and toxicological effects remains limited and controversial in the scientific literature [80]. Many organic compounds containing Te have been synthesized and investigated, revealing diverse pharmacological properties including neuroprotection, hepatoprotection, chemoprevention, and immuno-modulation, among others [81,82]. However, there is a narrow range between beneficial and toxic doses, raising concerns about the safety profile of these compounds. These paradoxical effects arise from alterations in cellular redox states induced by the compounds and gene modulation [80]. Organo-tellurium compounds are known for exhibiting several pharmacological properties, and many studies have described the promising role of bulk-Te and nano-Te, including the following:Protective effects on the nervous system.Protection of the liver.Potential as chemo-preventive agents, which means they might help prevent the development of cancer [83].Modulation of or influence on the immune system [35,84].Use in chemotherapeutic drugs [85] and enhancement in the cytotoxic activity of chemotherapeutic drugs [83].Use in chemo-photothermal synergistic therapy for cancer and tumors [86].

### 5.2. Biomedical Applications

The biosynthesis of Te-NPs using green technology has attracted a growing amount of attention, especially in the biomedical sector [17,18,43,74]. The difference between an applied Te dosage with a helpful effect and that with a harmful one is small, making it crucial to determine safe and effective dosages. Safety assessments are essential to determine the potential risks and benefits of using organo-tellurium compounds in various applications [84]. Tellurium has a potent ability to inhibit the growth of *E. coli*. So, *E. coli* can be selected as a model organism to assess the antibacterial properties of Te-loaded fibers [87]. Before the discovery and widespread availability of antibiotics, Te was used as an antibacterial agent. Its properties, including its anti-leishmaniasis, anti-inflammatory, anti-atherosclerotic, and immuno-modulating effects, have been known for years [88]. 

Tellurium QDs and related nano-structures have many applications in biological detection systems as biosensors, as well as in labeling, imaging, and targeted drug delivery systems. They have also been tested for their antibacterial and antifungal properties. Furthermore, Te-NPs exhibit unique properties like decreasing lipids and antioxidants and scavenging free radicals [88]. In recent years, Te-NPs and QDs have become popular in studies. QDs are tiny, light-emitting, nearly spherical semiconductor nano-crystal particles that are smaller than 10 nm and contain approximately 200–10,000 atoms. In clinical trials, CdTe QDs were used in photothermal therapy for the diagnosis and treatment of some types of tumors [83]. Ultrathin tellurium nano-sheets can have anticancer effects due to the combination of the TE nano-sheets’ photothermal effects and chemotherapy [89]. Naderi and co-workers investigated the cytotoxicity of three types of QDs (i.e., high-yield CdTe QDs, CdTe QDs, and CdTe/CdS core/shell QDs) on two human breast cancer cell lines. All three QD types induced apoptosis in a dose-dependent manner, shedding light on their cytotoxic mechanisms and contributing to our understanding of their safe use in biomedical applications [90]. 

Aloe-vera-based Te-NPs have anticancer activities against melanoma cells. These amorphous NPs (ATCC CRL-1619) were effective at concentrations between 5 and 100 μg mL^−1^ after 72 h of exposure. The explanation for this involves the elevation of reactive oxygen species (ROS) levels, which resulted in serious damage to the nucleic acids, lipids, and proteins at the molecular level as well as membranes and organelles at the macro level within the melanoma cell. At higher concentrations, powerful oxidants can damage vital macromolecules throughout the membranes and cytoplasm [83]. In another study, the cytotoxic activity of inorganic tellurite was compared with myco-synthesized Te-NPs as potential treatments for breast cancer. The Te-NPs provided great selective cytotoxic activity against a breast cancer MCF-7 cell line. At the same time, they had no cytotoxic effect against normal L929 cell lines at concentrations of 50 μg mL^−1^ or less. Potassium tellurite (K_2_TeO_3_) provided a higher level of cytotoxicity against the normal L929 cell line with an IC_50_ value of 76.33 μM (which corresponds to 9.739 μg mL^−1^ of elemental Te), and no IC_50_ value was found for the breast cancer MCF-7 cell line at concentrations of 100 μM or less (which correspond to 12.76 μg mL^−1^ of elemental Te). Moreover, biogenic nano-structured Te-NPs showed higher antioxidant and anticancer activities than potassium tellurite [83]. Bifunctional Te nano-dots with well-defined nano-structures have been developed as an effective cancer therapy. These nano-dots can be used with near-infrared light to induce hyperthermia within tumors (photothermal therapy) and generate ROS (photodynamic therapy) to damage cancer cells. They are encapsulated within hollow albumin nano-cages for targeted drug delivery. Te nano-dots show preferential accumulation in tumors and efficient renal elimination, reducing toxicity [91].

## 6. Toxicity of Tellurium and Nano-Tellurium

Tellurium is a highly toxic element to humans and microorganisms at low concentrations, mainly in its soluble oxy-anionic forms [10]. There are many differences between the bulk- and nano-forms of Te, which are summarized in Table 8. Important questions that need to be addressed include the following: (a) What are the toxic compounds that result from bulk and nano-tellurium? (b) What are the primary human toxicity symptoms from Te and Te-NPs? (c) What are the LD_50_ levels for Te and various nano-Te forms? (d) What is the metabolic pathway of Te in humans? And (e) what are the technological methods and industrial safety and health standards that are necessary to ensure our ability to work safely with Te?

### 6.1. Clinical Signs

Bluish-black skin discoloration on the face, neck, and webs of the fingers has been reported in connection with Te toxification [95]. Ingesting significant amounts of Te causes symptoms like vomiting, nausea, a metallic taste in the mouth, black discoloration of the oral mucosa and skin, corrosive injury to the gastrointestinal tract, and a distinctive garlic-like odor on the breath. The LD_50_ denotes the dose that causes the death of 50% of a test population [96]. In the case of elemental Te, the oral LD_50_ is >5000 mg kg^−^^1^ in rats [97]. For tellurium-dioxide (TeO_2_), the LD_50_ is >2000 mg kg^−^^1^ in rats [98]. Najimi and co-workers [92] compared the toxicity of biogenic tellurium nano-rods (Te-NRs) and potassium tellurite (K_2_TeO_3_) in mice. The LD_50_ value for Te-NRs was 60 mg kg^−^^1^, and for K_2_TeO_3_, it was 12.5 mg kg^−^^1^, indicating the biogenic Te-NRs were less toxic. Additionally, the “no-observed-adverse-effect level” dose for Te NRs in a 14-day subacute toxicity study was determined to be lower than 1.2 mg kg^−^^1^ [92]. Long-term Te poisoning can damage the nervous system, including peripheral neuropathy characterized by segmental demyelination and minor axonal degeneration. In the brain, Te particles cause changes that are localized in lipofuscin granules in the neuron cytoplasm. In the liver, Te can cause fatty degeneration and necrosis. In the kidneys, proximal tubular lesions, oliguria, or anuria have been reported as a result of Te poisoning. In the heart, Te can cause cell necrosis, edema, and congestion. Furthermore, reproductive effects, hydrocephalus, edema, exophthalmia, and ocular hemorrhage have been documented. These findings underscore the toxicological risks associated with tellurium exposure across various bodily systems and organs [99]. Physiologically based extraction tests showed that significantly more Te dissolves in synthetic stomach fluids compared to lung fluids, with its gastric bioaccessibility ranging from 13 to 31% of the total Te. This suggests a relatively low to moderate level of bioaccessibility, a common characteristic of elements associated with iron (oxy)hydroxides. These findings indicate that the Te present in tailings poses a moderate health concern [100]. Long-term studies on rats and mice exposed to Te in drinking water did not reveal any evidence of carcinogenic effects [101]. Organo-tellurium compounds can change enzyme levels in various organs, including the liver, kidney, brain, and serum. In the case of rats, Te can reduce levels of acetylcholine esterase and monoamine oxidase in serum and the brain, decrease hepatic glutathione levels and the levels of glutathione-S-transferase in the liver, and reduce alkaline phosphatase levels in the liver and kidney [102]. Tellurium compounds affect numerous enzymes in animals, leading to the demyelination of peripheral nerves and cell death via apoptosis. These compounds also impact enzymes in the glutamatergic system, causing damage via oxidative stress.

### 6.2. Biogeochemistry 

A good understanding of the biogeochemistry of Te is necessary within environmental studies, which may focus on Te sources, outcroppings of Te deposits, the geochemical exploration and extraction of Te, and the quantification of the environmental risks associated with its rapidly increasing anthropogenic usage [54]. These geochemical, anthropogenic, and biogeochemical processes are the main factors that control the mobility and distribution of Te in different environments [54]. Metabolically, Te is reduced in the liver, methylated into dimethyl tellurium and trimethyl tellurium, and eventually excreted via one’s urine and breath as volatile compounds [35]. Many studies have investigated the incorporation of Te into proteins and the possible biological role of chalcogenide metal nano-crystals derived from amino acid- and protein-bound Te forms, analogous to those of other sulfur-binding elements [77,103,104,105,106]. Organo-tellurium compounds have antioxidative, protective, or antiproliferative attributes in the cell system [107,108,109]. These organic Te compounds include hydroxyphenyl-telluride, aminophenyl-telluride, carboxyphenyl-telluride [107], diphenyl ditelluride [110], and thiophene-based organo-tellurium [109]. These organic compounds can protect kidney tissues against oxidative damage caused by reoxygenation and anoxia [107] due to their functional roles in therapeutics and clinical biology [82], as well as their pharmaceutical characteristics [109].

Ascorbic acid can mitigate the symptoms of Te poisoning, because it can reduce oxidized Te back to its elemental form (Te^0^) before the Te enters the biological pathway for methylation. Ascorbic acid was employed in the treatment of industrial workers exposed to Te-containing dust who had garlic-smelling breath. The garlic-like odor can result from the volatilization of (CH_3_)_2_Te when this compound leaves the body via one’s breath and urine. The metabolic pathway of Te takes a similar pathway as that of selenium and sulfur in both plants and humans, which is considered a process of detoxification by Te (Figure 5) [35,36,111]. This metabolism mainly depends on converting TeO_3_^2−^ into organo-Te compounds via certain redox enzymes, including glutathione reductase and thioredoxin reductase. Glutathione is the reducing agent that takes part in reducing TeO_3_^2−^ to elemental Te (Te^0^) accompanied by the formation of ROS [37]. Higher plants (e.g., garlic) can biologically generate Te-NPs when exposed to high concentrations (>1 mM) of soluble Te oxyanions [111]. Tellurium poisoning is not common because it is one of the rarest elements on Earth. Tellurium intoxication can cause circulatory collapse and respiratory depression. Moderate exposure can result in anorexia, the suppression of sweating, dry mouth, a metallic taste in one’s mouth, and a garlic odor in one’s breath and urine [112]. While tellurium might not initially seem biologically significant, it is present in human tissues, including the liver, bone, kidneys, lungs, ovaries, spleen, and testes, despite its rarity in Earth’s crust. This has prompted further investigation of its origins, abundance in soils, role as a constant component in biological materials, biological activity or inertness, and potential accumulation in mammalian tissues with age, as well as whether human exposure to Te is a natural occurrence or a result of industrial contamination [113].

### 6.3. Environmental Effects

The toxicology of Te also depends on environmental conditions, which control its fate and behavior. These environmental factors work along with human activities, especially industrial activities. Industrial applications of tellurium can endanger the environment and national security [114]. Due to increasing global demand for Te, environmental issues surrounding this metal have recently aroused concern [12]. Tellurium transformations are driven by microbes, including Te uptake, efflux, and reduction as well as the production of some biological Te-NPs. The fate of Te and its impact on a variety of environments has been studied [28,34,115,116,117]. Environmental factors that control the transformation and cycling of Te include conditions in the weathering zone, residual organic materials, climatic elements (temperature, humidity, etc.), and biological activity [93].

### 6.4. Toxicity of Nano-Tellurium

The cytotoxicity and carcinogenic effects of cadmium telluride QDs in normal human bronchial epithelial cells (BEAS-2B) depends on the particle size. In a study by Zheng et al. [118], three different CdTe QD particle sizes (mean diameters of 2.04 nm, 3.24 nm, and 5.40 nm, respectively) were investigated. When the BEAS-2B cells were acutely exposed to 2.04 and ~3.24 nm CdTe QDs, it resulted in dose-dependent cytotoxicity. Conversely, acute exposure to 5.40 nm CdTe QDs had minimal cytotoxic effects on the BEAS-2B cells. Remarkably, when the BEAS-2B cells were chronically exposed to CdTe QDs of all three tested particle sizes, it led to cellular transformation. This transformation was evidenced by an increase in cell migration and the ability of cells to grow independently in an anchorage-free environment, as observed in soft agar. These findings strongly suggest that some CdTe QDs possess the potential to act as potent carcinogens in human lung cells, specifically in BEAS-2B cells [118].

The toxicity of Te-NPs can be different. Sulphide/cadmium telluride (CdS/CdTe) QDs show size- and concentration-dependent toxicity. Larger aggregates of CdS/CdTe QDs (with sizes ranging from 25 to 100 nm) have a more pronounced inhibitory effect on phagocytosis compared to smaller nano-particles (those with sizes < 25 nm). Furthermore, different species exhibited distinct responses to these QD fractions [119]. Cadmium telluride QDs have the ability to transfer energy to adjacent oxygen molecules, initiating the generation of ROS. These ROS can subsequently trigger cellular inflammation, inflict damage, and ultimately result in cell death. Several studies have confirmed that CdTe QDs induce cell death in human neuroblastoma cells through mechanisms involving the upregulation of fats and lipid peroxidation [120]. High concentrations of CdTe-QDs exhibited cytotoxicity, evident through decreased cell metabolism and alterations in cell structures. In contrast, when exposed to sub-cytotoxic concentrations of CdTe-QDs, there was a notable reduction in the production of TNF-α, NO, KC/CXCL-1, and IL-8 in response to PA01 exposure, leading to the suppression of these cells. Prior exposure to CdTe-QDs could potentially hinder the targeted cells’ metabolism and weaken immune responses to bacteria, thereby increasing the host’s vulnerability to infection [121].

The toxicity of nano-tellurium compared to that of the bulk element is an important question that needs additional research. Nano-particles can become embedded in living tissue, such as the kidneys and liver. These NPs may act as surface catalysts for electron transfer and induce oxidative stress [122]. This has been confirmed for the application of nano-particles as a drug delivery system in biological systems [123]. Biocompatible Te nano-needles have been used for accelerated wound healing with a long-term stable antibacterial activity [124]. Many applications of nano-tellurium have been reported, as mentioned previously in this review, in the field of nano-medicine [11,13,74,125].

### 6.5. Tellurium Toxicology and Safety

Several metals/metalloids (like Te) are toxic to humans. Exposure can occur through environmental pollution [126], occupational sources [127], the water cycle, and the food chain [128,129]. Most of these metals/metalloids accumulate in organs upon chronic exposure [130] and their biological effects may contribute to the shortening of the organs’ lifespan, mainly the kidneys, and ultimately lead to their early failure [103]. Due to the toxicity of Te, industrial safety and health methods that reduce occupational exposure when Te is present are needed. It is likely that Te causes occupational disease (Figure 6) based on findings from studies such as [112,118]. Occupational diseases are any disease contracted as a result of exposure to risk factors arising from work activities, including work in factories, labs, and farms [131]. Thus, there is an urgent need to assess and manage occupational and ergonomic risks that may arise from Te exposure [132]. These studies should focus on safety and management in the work environment [133].

## 7. Conclusions

This review sought to answer several questions about Te, including (a) its role in modern industry, (b) the differences between bulk- and nano-tellurium, (c) the potential health impacts from Te exposure, and (d) the biochemical pathways through which Te influences living organisms. Tellurium is a rare element in Earth’s crust that was discovered in the eighteenth century. As a chalcogen, Te- and Te-NPs have great potential in photothermal, photoconductive, and electronic applications. However, the biological interactions of Te and its potential environmental and health impacts are not well understood. Tellurium and its nano-varieties have shown promise as anticancer, antibacterial, and imaging agents. However, the difference between potentially beneficial and toxic levels of Te is very small. Tellurium toxicity in plants and humans is likely caused by biochemical reactions that lead to the inactivation of proteins, but the biological role of bulk- and nano-Te in living organisms still needs further investigation. The main conclusion of this review is that both bulk- and nano-tellurium are promising agents that can be applied in the biomedical sector, but this and other uses of Te need additional investigations.

## Figures and Tables

**Figure 1 nanomaterials-14-00670-f001:**
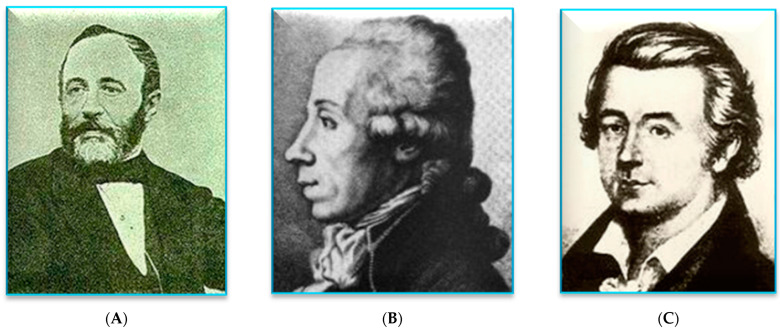
(**A**) Franz-Joseph Müller, (**B**) Martin Heinrich Klaproth, and (**C**) Pál Kitaibel. Sources: (**A**) [22], (**B**) [23], and (**C**) [24].

**Figure 2 nanomaterials-14-00670-f002:**
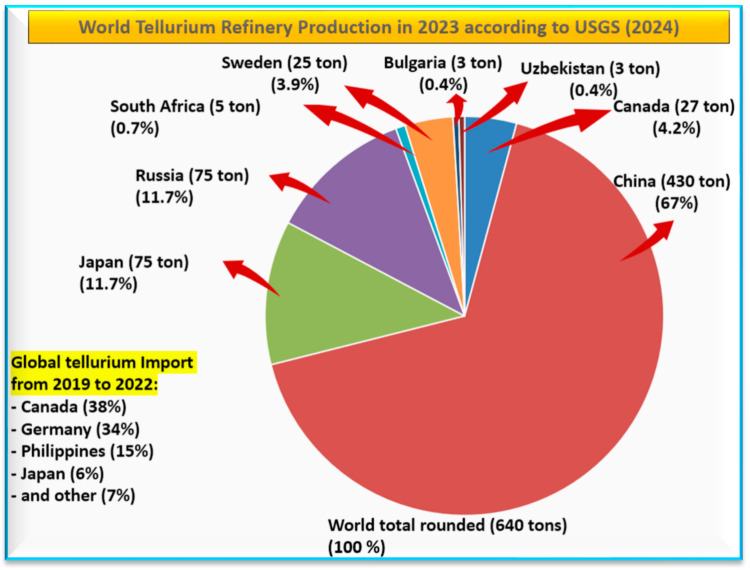
The main producers of tellurium (refinery production in 2023) and global import sources from 2019 to 2022 [30].

**Figure 3 nanomaterials-14-00670-f003:**
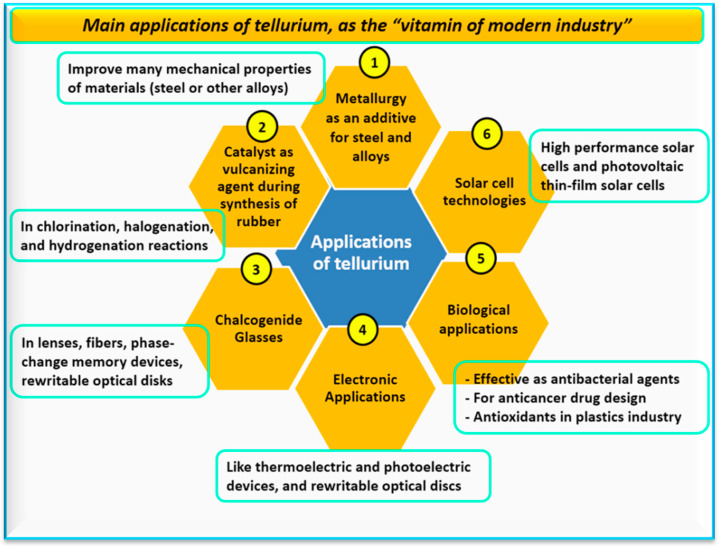
The main applications of tellurium as reported in the literature.

**Figure 4 nanomaterials-14-00670-f004:**
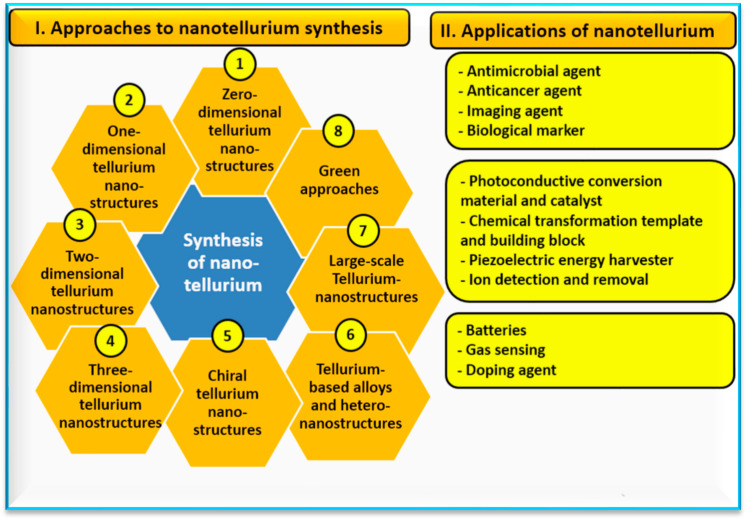
Ways to synthesize nano-tellurium (part I) and suggested applications (part II) [13,38].

**Figure 5 nanomaterials-14-00670-f005:**
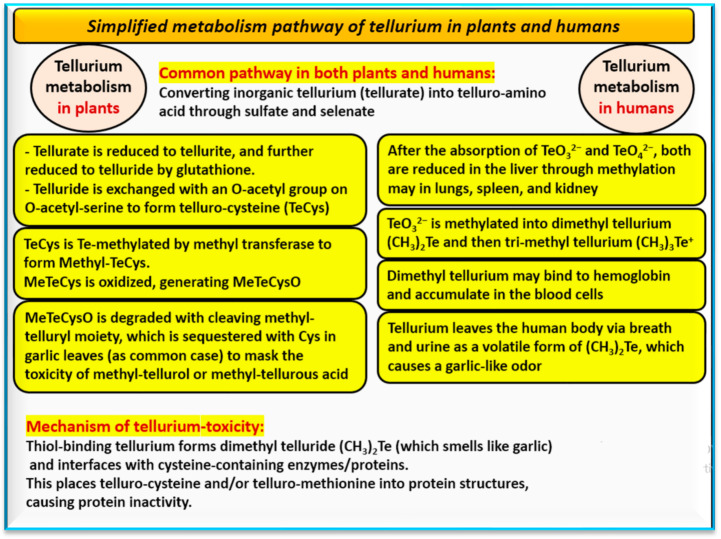
Simplified metabolism pathway of tellurium in plants and humans. The suggested mechanism of tellurium toxicity is shown as well [35,36,37,111].

**Figure 6 nanomaterials-14-00670-f006:**
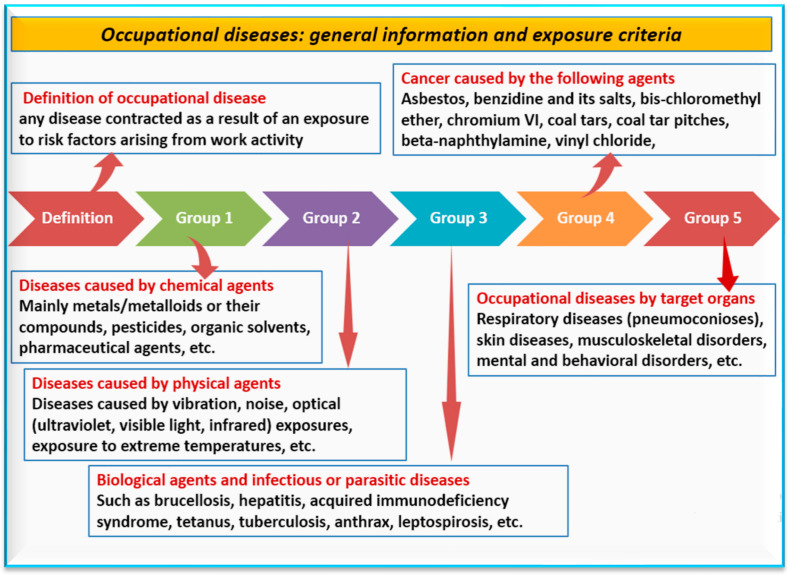
General information on occupational diseases and their causes [131]. Occupational diseases caused by tellurium would primarily fall into group 1.

**Table 1 nanomaterials-14-00670-t001:** Definitions for important terms related to nano-tellurium that are mentioned in this review.

Term(s)	Definition or More Details
Nano-materials (NMs)	A material in which at least one dimension of measurement is between 1 and 100 nm.
Nano-tellurium	Nano-tellurium or elemental Te-nano-particles (Te-NPs) is the nano form (diameter ranges from 1 to 100 nm) of the Te metalloid that can be found naturally in regolith samples produced by chemical or biological approaches.
Te-NP synonyms	Tellurium nano-powder, nano-tellurium, nano-Te
Natural nano-materials	A nano-material made naturally through (bio)geochemical or mechanical processes, without a direct or indirect connection to a human activity or anthropogenic process
Engineered nano-materials	Chemical substances/materials that are purposely created by humans with particle sizes between 1 and 100 nm in at least one dimension
Nano-alloy of tellurium	Produced by combining Te with other elements to create desirable properties. Common examples include combining Te with Se using microbes (e.g., *Lactobacillus casei* NCAIM B 1147), Pd–Te nano-clusters, CdTe QDs, CdTe/CdS QD nano-sensors, etc.
Common tellurium nano-materials	Nano-Te structures, nano-Te wires, nano-Te tubes, nano-Te rods, and nano-Te ribbons
Quantum dots (QDs)	Quantum dots (QDs) are semi-conducting nano-crystals with unique optical properties.
Cadmium telluride quantum dots (CdTe QDs)	CdTe-based quantum dots (QDs) are colloidal structures that have unique luminescence and electronic properties and are used in diagnostic and biomedical research. CdTe QDs have a low excitation energy and small band gap compared to those of CdS and CdSe materials.
Nano-needles	Needles in the nano-size range, used to deliver therapeutics intracellularly.
Nano-Te structures	Nano-forms of Te compounds that have a unique van der Waals structure and intriguing chemical and physical properties, such as nano-wires, nano-tubes, nano-cables, belt-shaped structures, etc.
Nano-particles (NPs)	Individual particles that range from 1 to 100 nm in diameter. Also sometimes used for nano-wires and nano-tubes.
Nano-spheres	Spherical particles with a diameter between 1 and 100 nm.
Nano-wires (NWs)	Nano-wires are structures with a diameter of ∼10 nm and a much greater length.
Nano-rods (NRs)	Nano-rods have a typical nano-size between 1 and 100 nm with standard aspect ratios (length divided by width) of 3–5.
Nano-tubes (NTs)	Nano-tubes are NMs with a cylindrical shape around a hollow center; their diameters typically range from 200 to 600 nm and their lengths from 4 to 15 nm.
Nano-ribbons (NRs)	Nano-ribbons are rectangular in shape, very thin with an appreciably greater width, and a length that can be hundreds of nm. Two dimensions are less than 100 nm.
Nano-plates	Nano-plates are rectangular in shape with only one dimension in the nano-meter range.
Dimensional Te nano-structures	NMs come in a variety of dimensions. Nano-particles are considered zero-dimensional (0D); nano-rods, nano-tubes, and nano-wires are one-dimensional (1D); nano-ribbons are two-dimensional (2D); and flower-like three-dimensional (3D) Te NMs have been created.

**Table 2 nanomaterials-14-00670-t002:** The main physical properties of tellurium metalloid [7,38].

Physical Parameter(s)	Value
Density in solid form at room T	6.0 (amorphous) and 6.25 (crystalline) g cm^−3^
Density in liquid form	5.70 g cm^−3^
Thermal conductivity	1.0–3.4 W/(m·K) in a single crystal
Electrical resistivity	1–50 mΩ·m
Electron affinity	1.971 eV or 1.96 eV
Electronegativity	2.1 (Pauling scale), 2.158 (Allen scale)
Van der Waals radius	206 pm
Ionization energy	9.010 eV
Band gap energy	0.35 eV at room temperature
Molar volume	2.05 × 10^−5^ m^3^
Crystal structure	Trigonal, orthorhombic, or hexagonal form
Atomic radius	123 pm
Covalent radius	138 pm
Atomic number	52
Melting point	449.5 °C (amorphous) and 452 °C (crystal)
Boiling point	988 °C (amorphous) and 1390 °C (crystalline)
Heat of fusion	17.5 kJ mol^−1^
Heat of vaporization	48 kJ mol^−1^
Specific heat	199–219 J/(kg.K)
Refractive index	1.002495 at λ = 589 nm (vapor) 1.26 at 260 nm and 4.5 at 720 nm (thin film)

**Table 3 nanomaterials-14-00670-t003:** The main chemical properties of tellurium metalloid [7,38].

Chemical Parameter(s)	Details
Oxidation states	Telluride (Te^2−^) (−2), elemental (Te^0^) (0), tellurite (TeO_3_^2−^) (+4), and tellurate (TeO_4_^2−^) (+6). Tellurite (TeO_3_^2−^) (+4) is the most common.
Common Te alloys	Bismuth telluride (Bi_2_Te_3_) with Se or Sb, mercury cadmium telluride (HgCdTe), cadmium zinc telluride (CdZnTe), etc.
Common Te minerals	Calaverite (AuTe_2_), sylvanite (AgAuTe_4_), krennerite (AuTe_2_), nagyagite [AuPb(Sb, Bi)Te_2_–3S_6_], and tellurobismuthite (Bi_2_Te_3_)
Main Te-oxides in crystalline, amorphous, and colloidal forms	Telluride (Te^2−^), tellurite (TeO_3_^2−^), tellurate (TeO_4_^−^), tellurium dioxide (TeO_2_), and tellurium trioxide (TeO_3_)
Tellurium hydrides	Hydrogen telluride (H_2_Te), hydrogen-rich varieties such as H_4_Te, and H_5_Te_2_
Tellurium acids	Tellurous acid (H_2_TeO_3_) and telluric acid [Te(OH)_6_]
Tellurium electron conifiguration	[Kr] 5s^2^ 4d^10^ 5p^4^
Organo-tellurium compounds	Functional group –TeH is called tellurols and they are found in compounds such as dimethyl telluride (Te(CH_3_)_2_) and diphenyl ditelluride (C_12_H_10_Te_2_)
Isotopes of tellurium	Stable forms are ^120^Te, ^124^Te, ^125^Te, and ^126^Te
Common Te crystal morphologies	n-whiskers, f-whiskers, s-whiskers, etc.

**Table 4 nanomaterials-14-00670-t004:** Chemical characterization of selected tellurium compounds found with 2D structure [7].

Te Compound (s)	Molecular Formula	Molecular Weight	Synonyms
Tellurous acid	H_2_O_3_Te	177.6 g/mol	Dihydrogen trioxotellurate
Tellurinic acid	H_2_O_2_Te	161.6 g/mol	Hydridohydroxidooxidotellurium
Tellane	H_2_Te	129.6 g/mol	Hydrogen telluride
Tellurium, mol.	Te2	255.2 g/mol	Di tellurium
Copper telluride	CuTe	191.1 g/mol	Copper monotelluride
Copper tellurate	CuO_4_Te	255.1 g/mol	Copper(2+) tellurium tetraoxide
Cuprous telluride	Cu_2_Te	254.7 g/mol	Copper (I) telluride
Tellurium trioxide	TeO_3_	175.6 g/mol	Tellurium(VI) oxide
Tellurium dioxide	TeO_2_	159.6 g/mol	Tellurium oxide
Cadmium telluride	CdTe	240.0 g/mol	Cadmium monotelluride
Cadmium tellurate	CdTeO_4_	304.0 g/mol	Cadmium tellurium tetraoxide
Bismuth telluride	Bi_2_Te_3_	800.8 g/mol	Bismuth selenide telluride
Mercury telluride	HgTe	328.2 g/mol	Tellanylidenemercury
Dimethyl telluride	C_2_H_6_Te	157.7 g/mol	Dimethyltellurium
Diphenyl ditelluride	C_12_H_10_Te_2_	409.4 g/mol	Ditelluride, diphenyl
Potassium tellurite	K_2_TeO_3_	253.8 g/mol	Dipotassium trioxotellurate
Potassium tellurate	K_2_TeO_4_	269.8 g/mol	Potassium tellurate(VI)
Sodium tellurite	Na_2_TeO_3_	221.6 g/mol	Disodium trioxotellurate
Sodium tellurate	Na_2_TeO_4_	237.6 g/mol	Disodium tetraoxotellurate
Tellurium hexafluoride	F_6_Te	241.6 g/mol	Tellurium(VI) fluoride
Phenol, 4,4′-tellurobis-	C_12_H_10_O_2_Te	313.8 g/mol	bis(4-hydroxyphenyl)telluride
Diphenyl ditelluride	C_12_H_10_Te_2_	409.4 g/mol	Phenyl ditelluride

**Table 5 nanomaterials-14-00670-t005:** Some published studies on biological and green production of nano-Te.

Synthesis Type	Produced by	Nanparticle Features	Main Purpose of the Study	Refs.
Biosynthesis	*Streptomyces graminisoli.*	Crystal shape (12–25 nm)	Antibacterial activity; minimum inhibitory concentration was 50 μg mL^−1^	[11]
Biogenic method	*Penicillium chrysogenum* PTCC 5031	50.16 nm	Exploited biomolecules and enzymes secreted from *P. chrysogenum* at room temperature	[42]
Biosynthesis of nano-Te rods	*Gayadomonas* sp. TNPM15	15–23 nm	Acted against phytopathogenic fungi by disruption of integrity and membrane permeability of fungal spores	[18]
Biogenic Te-NPs	Bacterial marine isolates	Smaller than 100 nm	Antimicrobial activity	[43]
Biosynthesized Te-NPs	*Lysinibacillus* sp. EBL303.	Rod-shaped (22–148 nm)	Bioremediation of tellurite and phenol at polluted sites	[44]
Tellurium nano-rods	*Shewanella baltica*	From 8–75 nm	Reduced methylene blue through photo-catalytic and anti-biofilm activity	[45]
Biogenetic nano-Te particles	*Mortierella* sp. AB1	From 100–500 nm	Antibacterial against *Escherichia coli*, *Shigella dysenteriae*, *Salmonella typhimurium*, and *Enterobacter sakazakii*	[46]
Biogenic Te-NPs.	*Aspergillus welwitschiae*	Spherical shape (60.80 nm)	Antibacterial activity against *Staphylococcus aureus* and *E. coli*	[47]
Biosynthesis of Te-NPs	Biomolecules of gallic acid	19.74 nm	Multifunctional agents and biomedical applications	[48]
Green synthesis of Te-NPs	*Allium sativum* extract	350 nm	Evaluation of the cytoprotective and antioxidant activities of Co-Te-NPs	[49]

**Table 6 nanomaterials-14-00670-t006:** Common methods to synthesize nano-Te structures.

Nano-Te Structures	Common Form	Method of Synthesis	Additional Details
Zero-dimensional Te nano-structures	NPs in a spherical morphology	Green and chemical synthesis or laser ablation in liquids	Sizes of produced Te-NPs depend on solvents used
One-dimensional (1D) Te nano-structures	Nano-wires (Te-NWs)	Microwave-assisted synthesis, hydrothermal methods, and vapor–solid method	Te-NWs are controlled by the temperature of reaction, substrate, and growth time
	Nano-tubes (Te-NTs)	Physical vapor deposition	Te-NTs are controlled by substrate and deposition temperature
	Nano-ribbons (Te-NRs)	Hydrothermal and vapor deposition methods	Te-NRs are controlled by pH, temperature, and reaction time
	Tellurium nano-rods	Hydrothermal methods	Surfactants can control diameters and lengths of nano-Te rods
	Tellurium belt-shaped structures (Te-NPs)	Thermal evaporation and deposition methods	Temperature and ambient atmosphere control NPs
Two-dimensional (2D) nano-Te structure	2D tellurene lesser layers	Physical vapor deposition and liquid-phase evolution	Temperature and thermodynamics control NPs
Three-dimensional (3D) nano-Te structure	Flower-like 3D Te nano-structures	Solvothermal method, dissolution, and recrystallization	Type of solvents (e.g., water, amide, or alcohol) and temperatures control NPs
Chiral-shaped Te nano-structures	Chiral nano-materials (NMs)	Using chiral biomolecules as initiators	Chiral NMs are controlled by different synthetic conditions

**Table 7 nanomaterials-14-00670-t007:** Tellurium alloys or chemical compounds used in the energy sector.

Te-Alloys	Molecular Formula	Main Findings	Refs.
Alloys from sodium, yttrium, sulfur, and tellurium	NaYS_2(1−*x*)_Te_2*x*_ alloys (with *x* = 0, 0.33, 0.67, and 1)	These alloys are potential light energy converters and considered attractive for photovoltaic applications	[64]
Alloys of tellurium fluorides	Te-F binary system	Stable Te-fluorides (TeF_4_, TeF_6_, and TeF_8_,) support strong *d*–*p* covalent interactions in the Te-F system at high pressure	[65]
Iridium–tellurium alloy	IrTe	Can promote adsorption of N_2_ and lower the Gibbs free energy for electrocatalytic N_2_ reduction reactions	[66]
Bismuth–telluride alloy	Bi_2_Te_3_	This alloy has good performance in thermoelectric materials near room temperature	[67]
Cesium–tellurium–titanium alloy	Cs_2_Te_1−*x*_Ti*_x_*I_6_	This alloy possesses large absorption coefficients in the visible light region as stable, eco-friendly and high-efficiency light absorbers used in optoelectronic applications	[68]
Rubidium–tin–tellurium alloy	Rb_2_Sn_1−*x*_Te*_x_*I_6_	Promising alloy using the Sn–Te mixture as a potential substitute for lead in photovoltaic materials	[69]
Tellurium-embedded carbon nano-fibers	Te@C-NF electrode	Poly-tellurides and K_2_Te-embedded carbon nano-fibers are high-rate and long-life electrodes for high-energy-storage materials	[70]
Potassium–tellurium battery system	K-Te	Converting Te to K_2_Te_3_ and ultimately to K_5_Te_3_ in a carbonate electrolyte-based K-Te battery system to promote and develop high-energy-density K-S/Se/Te batteries	[71]
Potassium–tellurium battery system	K-Te	Utilizes biochar from mangosteen shell in a hierarchical porous host to Te during K^+^ storage in K-Te battery	[72]
Amorphous selenium (a-Se)–tellurium alloy	Se-Te alloys	Improving quantum efficiency and conversion efficiencies for a-Se_1−*x*_Te*_x_* (*x* = 0, 0.03, 0.05, 0.08) devices as a function of applied field, along with different band gaps in Se-Te alloys	[73]

**Table 8 nanomaterials-14-00670-t008:** A comparison between tellurium and nano-tellurium [10,12,35,37,83,92,93,94].

Comparison Item	Bulk-Tellurium	Nano-Tellurium
Main common forms	Soluble oxyanionic forms	Natural and engineered nano-particles (NPs)
Abundance	1–5 ppb in Earth’s crust 0.008–0.03 ppm in soil 15 ppb in seawater Around 0.27 ppm in plants	>4 ppm in the regolith depending on weathering of Te-ores >100 ppm in hotspots
Essentiality	Non-essential	Not confirmed yet
Exposure pathway(s)	Food (ingestion) followed by inhalation and then dermal.	Bioavailability of Te-NPs through dermal absorption, ingestion, or inhalation.
Foodstuff exposure and human daily intake	Dairy products, meat, and cereals; in general, there is <1 mg Te kg^−1^ in food and humans should not exceed an intake of >0.1 mg of Te day^−1^	Depends on natural or engineered NPs and their properties
Main sources of exposure	Mainly Cd-Te in solar panels and from copper mining refineries	Cd-Te-quantum dots (QDs) and other nano-alloys of tellurium
Main applications	Te can be used as an alloy for Peltier devices, phase change optical magnetic disks, and solar panels	Alloys of Te with selenium, cadmium, zinc, and other metals can be used to produce NMs such as QDs
Mobility in the environment	Tellurium is a mobile element in the environment (mainly mining)	Te-NPs may transport similarly to other natural nano-materials like Au-NPs
Suggestied mechanisms to enhance human health	Boosts antioxidant defenses, acts as pro-oxidants, generates ROS, and induces apoptosis	Exerts antioxidant, lipid-lowering, and free radical scavenging activities; can be used as antitumor and chemopreventive agents
Toxicity (established)	Low concentrations of Te species are toxic	Elemental (Te^0^) is non-toxic to organisms
Toxicity (exposure dose)	Te^IV^O_3_^2^-_(aq)_ toxic to microbes at about ~1 mg L^−1^ (4 μM)	Bio-Te-NPs caused toxicity to *Pseudomonas pseudoalcaligenes* in mice at 6 mg kg^−1^
Occupational exposure limits	Threshold limit value (TLV): 0.1 mg m^−3^ as 8 h total weighted average (TWA)	Not yet known
Toxicity (sources)	Highly toxic forms: tellurite, IV (TeO_3_^2−^) and tellurate, VI (TeO_6_^6−^)	Chemical Te-NPs are generally more toxic than biological or green forms
Toxicity (forms)	Organo-Te compounds are generally less toxic compared to mineral forms	Generally, the common toxic nano-form is Cd-Te QDs depending on size of NPs
Median lethal dose (LD_50_)—oral	K_2_TeO_3_ caused complete toxicity at 12.5 mg kg^−1^ in mice	Biogenic nano-Te rods had acute toxicity at 60 mg kg^−1^ in mice
Metabolic pathway	Reduces TeO_3_^2−^ and TeO_4_^2−^ in the liver, methylates to (CH_3_)_2_Te and (CH_3_)_3_Te)^+^, binds to hemoglobin, accumulates in the blood cells in humans	In general, bio-Te-NPs are insoluble in plants, depending on type of nano-tellurium (e.g., TeO_2_-NPs, TeO_2_-NP–acetic acid, and TeO_2_-NP–gallic acid)

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
