# Peer review of "Tellurium and Nano-Tellurium: Medicine or Poison?"

_nanomaterials, 2024, doi:10.3390/nano14080670_

Round 1

Reviewer 1 Report

Comments and Suggestions for Authors

 Although it is an exhaustive study about this chalcogen element, the manuscript is quite difficult to read because the information presented is not very rigorously systematized and the wording of the phrases is quite cumbersome. I noticed several mistakes in expression and technical editing, which I pointed out directly in the manuscript with track changes.

I believe the manuscript matched the scientific scope of NANOMATERIALS and can be considered for publication after major  corrections.

Comments on the Quality of English Language

I noticed several mistakes in expression and technical editing, which I pointed out directly in the manuscript with track changes.

Author Response

Response to Reviewer 1

Dear Reviewer,

Author Response: Many thanks for your time and your valuable comments.

Comments and Suggestions for Authors

Although it is an exhaustive study about this chalcogen element, the manuscript is quite difficult to read because the information presented is not very rigorously systematized and the wording of the phrases is quite cumbersome. I noticed several mistakes in expression and technical editing, which I pointed out directly in the manuscript with track changes. I believe the manuscript matched the scientific scope of NANOMATERIALS and can be considered for publication after major corrections.

Author Response: Many thanks for your time and your valuable comments.

We followed your comments and tried to answer as follows:

Reviewer comment (s)

Author response

Line 11:

please reformulate in agreement with AI information: The element was found in a gold ore from the mines in Kleinschlatten (today Zlatna), near the present-day city of Zlatna, Romania.

Modified and done based on the advice of the reviewer, thanks.

Line 13:

it is not correlated with the phrase.

This part was deleted, thanks.

Line 52:

That for, delete

Done, thanks

Line 69:

The

Done, thanks

Line 75-76:

“However, further research revealed that this element was actually a compound of gold and an unknown substance”

It is not clear the meaning of this sentence, please reformulate!

Changed into:

This statement has been modified, thanks.

Line 90:

Occurrence and forms of tellurium in nature; physical and chemical properties

Done, thanks

Line 115:

oxidic

Changed, thanks

Line 138

comma

Done, thanks

Line 147:

Delete “which was” and adding are

Done, thanks

Line 149:

Are presented

Done, thanks

Line 158:

tellurium metalloid

Done, thanks

Line 159:

Compound of “delete”

Done, thanks

Line 157:

generated

Done, thanks

Line 176:

in industrial and both health care applications

Done, thanks

Lines 209-210:

Naturally associated with the respiration of the microorganisms for instance Saccharomyces cerevisiae Te(0) can be raised, the fermentation can increase the production.

It is not clear the meaning of this sentence, please reformulate!

Corrected to:

Te(0) can be produced naturally depending on the respiration of the microorganisms (e.g., Saccharomyces cerevisiae), as the fermentation can increase this production.

Line 232:

delete

Done thanks

Line 240:

on the fact that tellurium is considered the

Done based on the reviewer comment, thanks

Line 287:

Production instead of producing

Done, thanks

Line 299:

that

Done, thanks

Line 358:

superscript

Done, thanks

Line 365:

Delete “the”

Done, thanks

Line 368:

Delete while

Done, thanks

Line 385:

Applied “delete”

Done, thanks

Line 441:

tellurium particles have been observed and

Done, thanks

Line 442:

Delete

Done, thanks

Line 445

Also affect

Done, thanks

Line 490:

aimed to answer several questions, some of them not completely,

line 491:

this element

Done, thanks

Line 502:

of this paper is

Done thanks

Comments on the Quality of English Language

I noticed several mistakes in expression and technical editing, which I pointed out directly in the manuscript with track changes.

Author Response: Many thanks for your comments, we went through the MS and followed your comments to improve the revised MS, thanks again. We also had a native English-speaking colleague review the manuscript and help improve the English.

Reviewer 2 Report

Comments and Suggestions for Authors

Dear Authors, 

Draft “nanomaterials-2914123-peer-review-v1 - Tellurium and Nano-Tellurium, medicine or poison?” addresses a niche, yet interesting issue in the toxicology of mineral elements, under a well-learned and unusual perspective that makes it suitable to the aims of MDPI-Nanomaterials. On reading, I find this review article instructive for myself, and hopefully for several other readers.

Comments as Reviewer. Just from the start, the Authors should revise the draft for minor but frequent mistyping and trivial errors, so that the final version will be smoother to read. Check that also the text in the Figures (they are mostly recycled ppts, I won’t agree that this makes good publication) is with good grammar and sense. Table 1 “Tellurium hylides” is it “hydrides”? Is there any Supplementary Materials? If not, please strike away the deceiving section.

Comments as reader: a review like this makes a good fact source reference for short occupational medicine chapters like those you can find in the recent ILO Criteria book (see and download freely at: https://www.ilo.org/global/topics/safety-and-health-at-work/resources-library/publications/WCMS_836359/lang--en/index.htm). It would be a bonus to readers to add, whether possible, a proportionate section on technological methods and industrial safety and health that would complete the survey on toxicology. The discussion on Te nanoparticle toxicity is interesting, but confusing. Several readers would appreciate if you restructure the text and separate clinical signs (ll 390-391; 425-428), biochemistry (make a good table of organic forms, such as using Metallomics 2013), environmental effects (organication by microorganisms). It would be an advantage to make suitable sub-chapters from the text blocks of Figure 7.

I have a comment on the possible biological role of chalcogenide-metal nanocrystals derived from amino-acid and protein-bound Te forms, analogous to those of other Sulfur-binding elements. (doi:10.3390/toxics3010020; it is pointless to cite it, however it has more than 200 refs) that you may see whether it fits into the general picture. Nanoparticles embedded in living tissue, such as the kidney and the liver, may act as surface catalysts for electron transfer and trigger ox-stress with mechanisms analogous to light-producing in nanoparticle-based LEDs. Free for you to discuss relevance in the frame of Te toxicology.

While I appreciate very much your draft, I believe that a good tier of revision is very necessary. 

Kind regards,

Comments on the Quality of English Language

Apart from typos and trivial mistakes, here and there, the text does not flow. While restructuring the text, you can make the mends.

Author Response

Response to Reviewer 2

Dear Reviewer,

Author Response: Many thanks for your time and your valuable comments.

Comments and Suggestions for Authors

Dear Authors,

Draft “nanomaterials-2914123-peer-review-v1 - Tellurium and Nano-Tellurium, medicine or poison?” addresses a niche, yet interesting issue in the toxicology of mineral elements, under a well-learned and unusual perspective that makes it suitable to the aims of MDPI-Nanomaterials. On reading, I find this review article instructive for myself, and hopefully for several other readers.

Author Response: Many thanks for your time and your valuable comments.

Many thanks for your kind words and encouragement.

Comments as Reviewer. Just from the start, the Authors should revise the draft for minor but frequent mistyping and trivial errors, so that the final version will be smoother to read.

Author Response: Many thanks for your time and your valuable comments.

The revised MS has been improved and double-checked. A native English-speaking colleague has reviewed the manuscript to improve the writing.

Check that also the text in the Figures (they are mostly recycled ppts, I won’t agree that this makes good publication) is with good grammar and sense.

Author Response: Many thanks for your time and your valuable comments. The figures have been reviewed.

Table 1 “Tellurium hylides” is it “hydrides”? Is there any Supplementary Materials? If not, please strike away the deceiving section.

Author Response: Many thanks for your comment,

We corrected and modified, thanks

Comments as reader: a review like this makes a good fact source reference for short occupational medicine chapters like those you can find in the recent ILO Criteria book (see and download freely at: https://www.ilo.org/global/topics/safety-and-health-at-work/resources-library/publications/WCMS_836359/lang--en/index.htm ). It would be a bonus to readers to add, whether possible, a proportionate section on technological methods and industrial safety and health that would complete the survey on toxicology.

Author Response: Many thanks for your time and your valuable comments.

Many thanks for your great book. This book is really very important, and we tried to pick up some important information that will be very useful to the readers. We added this section to the revised MS:

6.5 Tellurium toxicology and safety

The discussion on Te nanoparticle toxicity is interesting, but confusing. Several readers would appreciate if you restructure the text and separate clinical signs (ll 390-391; 425-428), biochemistry (make a good table of organic forms, such as using Metallomics 2013), environmental effects (organication by microorganisms). It would be an advantage to make suitable sub-chapters from the text blocks of Figure 7.

Author Response: Many thanks for your comment. We divided this section into your suggested sub-sections as follows:

  1. Toxicity of tellurium and nanotellurium

6.1 Clinical signs

6.2 Biogeochemistry

6.3 Environmental effects

6.4 Toxicity of nano-tellurium

6.5 Tellurium toxicology and safety

We will make a table as suggested for our next review on nanotellurium. Due to the short time for corrections given by this journal we do not have time to construct the table, but we added a information on organotellurium compounds in the Biogeochemistry section.

Furthermore, we are working right now in the lab and have obtained interesting results on nano-tellurium and its alloys with nanoselenium, which we published in the following article:

Muthu, A.; Sári, D.; Ferroudj, A.; El-Ramady, H.; Béni, Á.; Badgar, K.; Prokisch, J. Microbial-Based Biotechnology: Production and Evaluation of Selenium-Tellurium Nanoalloys. Appl. Sci. 2023, 13, 11733. https://doi.org/10.3390/app132111733

We also added a section on the biogeochemistry of tellurium in the revised MS.

I have a comment on the possible biological role of chalcogenide-metal nanocrystals derived from amino-acid and protein-bound Te forms, analogous to those of other Sulfur-binding elements. (doi:10.3390/toxics3010020; it is pointless to cite it, however it has more than 200 refs) that you may see whether it fits into the general picture.

Author Response: Many thanks for your time and your valuable comments.

We added section 6.5 to help address this.

Nanoparticles embedded in living tissue, such as the kidney and the liver, may act as surface catalysts for electron transfer and trigger ox-stress with mechanisms analogous to light-producing in nanoparticle-based LEDs. Free for you to discuss relevance in the frame of Te toxicology.

Author Response: Many thanks for your comment.

We added the following part to the revised MS:

Toxicity of nano-tellurium compared to the bulk element is an important question that needs additional research. Nanoparticles can become embedded in living tissue, such as the kidneys and liver. These NPs may act as surface catalysts for electron transfer and in-duced oxidative-stress [112]. This has been confirmed for the application of nanoparticles as a drug delivery system in biological systems [113]. Biocompatible Te nanoneedles have been used for accelerated wound healing under long-term stable antibacterial activity [114]. Many applications of nanotellurium have been reported, as mentioned previously in this review, in the field of nano-medicine [11] [13] [64] [115].

While I appreciate very much your draft, I believe that a good tier of revision is very necessary.

Author Response: Many thanks for your comments. We feel there are many improvements in our revised MS thanks to your comments, along with comments of other reviewers and the editor as well.

Kind regards,

Author Response: Many thanks for your time and your valuable comments. Definitely, these comments will improve our revised MS, thanks.

Comments on the Quality of English Language

Apart from typos and trivial mistakes, here and there, the text does not flow. While restructuring the text, you can make the mends.

Author Response: Many thanks for your comment. The text has been reviewed and improved by a native English-speaking colleague.

We hope our revised MS is improved and accepted by the reviewer.

Thanks, again

Round 2

Reviewer 1 Report

Comments and Suggestions for Authors

I agree with the paper publication in this form

Author Response

Many thanks for your time and encouragements

Reviewer 2 Report

Comments and Suggestions for Authors

Dear Authors, dear Editor,

The second-tier draft of “nanomaterials-2914123-peer-review-v1 - Tellurium and Nano-Tellurium, medicine or poison?” shows several improvements over the first one. However, I have much the same issues on the organization of statements throughout the text that make reading (I am reading both versions in print, rather than on screen, because scrolling up and down is even worse) rather dizzying. I have very little question on your abundant learning on this niche topic, however taking home the general points is far from easy. Please remind that efficient indexing will drive on this review many readers from disparate fields, even casual and just curious, so everyone should be able to find well organized relevant information without jumping up and down the text.

If ever I dare forward a suggestion, you may make an Introduction (why is it important, I should continue reading this and not move to the Wikipedia article) and general properties (including Table 1) section and then summarize the points you will address. Also useful is defining early terms such as nano particles, ribbons, rods and the like, so that readers won’t get confused. Another more substantial point is, throughout the text, separating the information on different chemical forms (Te(0), different physical forms like particles, wires, ribbons; different valence states and combinations with other elements, and so on). When discussing the toxicity of different nano-Te forms, such as “different CdTe QD particle sizes (520Q, 580Q, and 730Q, respectively)”, you may describe or define them before.

Another tier of contents classification is on biological forms and on their transformations, including conversion of organicated forms to Te(0) in its different physical forms. When discussing materials, please state clearly whether they are different nano-forms of elemental Te or of alloys and compounds (CdTe, and others). You may make a good discussion on whether they are alloys or chemical compounds, given the low energy differences between orbitals and the possibility that electron transfer can occur in the bulk of the nano-solids (you just made a teaser on “mixed valence” polymeric forms that is worthy of further enlightenment). These aspects make readers curious and in search of good discussion of complicated topics, if only to widen general knowledge.

There is a lot of confusion also in chapters 6.1 and 6.2 with toxicity and biogeochemistry intermingled among the two.

As you can understand, this is just a short list of suggestions that spring from several reading tiers of your second draft. Please, believe that I appreciate very much this nice compilation and highly esteem your efforts to make a point of more than 100 references on a niche topic. However, I still find the compilation, rather than the backgrounding science, still poorly organized. I would ask you to make the extra effort to rewrite a better draft, by re-organizing the contents in a smoother and better articulated way. It is a pity that this Journal does not allow Table of content in the reviews, because preparing that is very helpful in organizing the contents.

I again mark this draft for “major revision”, and I hope you understand the motivation: it is admirative, not despising! I strongly hope that you can make a much better, more informative review article to the benefit of many, many readers.

Kind regards.

Comments on the Quality of English Language

nothing more than check typos and go along with text revision.

Author Response

Response to Reviewer 2#

Dear Reviewer 2#

Many thanks for your time and comments.

We have worked to answer your comments. We added many new parts to the manuscript based on your advice. We believe your additions will improve our MS. We hope this round will get your final approval, thanks again.

Comments and Suggestions for Authors

Dear Authors, dear Editor,

The second-tier draft of “nanomaterials-2914123-peer-review-v1 - Tellurium and Nano-Tellurium, medicine or poison?” shows several improvements over the first one. However, I have much the same issues on the organization of statements throughout the text that make reading (I am reading both versions in print, rather than on screen, because scrolling up and down is even worse) rather dizzying. I have very little question on your abundant learning on this niche topic, however taking home the general points is far from easy. Please remind that efficient indexing will drive on this review many readers from disparate fields, even casual and just curious, so everyone should be able to find well organized relevant information without jumping up and down the text.

Response of author: Many thanks for your advice and comments. We made more improvements by adding new tables (for example, Table 1 now provides definitions for many nano-terms that the reader may not be familiar with) and modifications to some of the previous tables. We double checked our personal TOC for the MS again as your asked, and tried to improve readability of the revised MS.

If ever I dare forward a suggestion, you may make an Introduction (why is it important, I should continue reading this and not move to the Wikipedia article) and general properties (including Table 1) section and then summarize the points you will address.

Response: Thanks a lot for your comment.

Modifications have been made to the Introduction in an attempt to address the reviewer’s comments. See particularly the final paragraph of the introduction.

Also useful is defining early terms such as nano particles, ribbons, rods and the like, so that readers won’t get confused.

Response: Thanks a lot for your comment. We have added a new Table 1 in the revised MS. This table includes definitions and details for many items mentioned in the MS.

Another more substantial point is, throughout the text, separating the information on different chemical forms (Te(0), different physical forms like particles, wires, ribbons; different valence states and combinations with other elements, and so on).

Response: Thanks a lot for your comment.

Text was improved and more improvements were added in 3 new Tables (2, 3, and 4). These tables give summaries of information about Te properties and compounds.

When discussing the toxicity of different nano-Te forms, such as “different CdTe QD particle sizes (520Q, 580Q, and 730Q, respectively)”, you may describe or define them before.

Response: Thanks a lot for your comment. We have defined important terms in the new table 1 as previously mentioned.

Another tier of contents classification is on biological forms and on their transformations, including conversion of organicated forms to Te(0) in its different physical forms. When discussing materials, please state clearly whether they are different nano-forms of elemental Te or of alloys and compounds (CdTe, and others).

Response: Thanks a lot for your comment. We have added a new Table 1 and Table 4 in the revised MS that will prepare the reader to understand the terminology they encounter later in the manuscript.

You may make a good discussion on whether they are alloys or chemical compounds, given the low energy differences between orbitals and the possibility that electron transfer can occur in the bulk of the nano-solids (you just made a teaser on “mixed valence” polymeric forms that is worthy of further enlightenment). These aspects make readers curious and in search of good discussion of complicated topics, if only to widen general knowledge.

Response: Thanks a lot for your comment. We added a new Table 7 based on your advice to provide more explanations to the readers in the revised MS, thanks

New refs. used to create Table 7

The number in [ ] refers to the number of the reference in the manuscript:

[64] Azzouz L, Halit M, Charifi Z, Matta CF. Tellurium Doping and the Structural, Electronic, and Optical Properties of NaYS2(1-x)Te2x Alloys. ACS Omega. 2019 Jun 28;4(6):11320-11331. doi: 10.1021/acsomega.9b01330.

[65] Gao, Y.; Cui, T.; Li, D. Unexpected d−p Orbital Covalent Interaction Between the Non-d-block Main-group Metal Tellurium and Fluorine at High Pressure. Fundamental Research,

  1. https://doi.org/10.1016/j.fmre.2023.05.001.

[66] Jiang B, Xue H, Wang P, Du H, Kang Y, Zhao J, Wang S, Zhou W, Bian Z, Li H, Henzie J, Yamauchi Y. Noble-Metal-Metalloid Alloy Architectures: Mesoporous Amorphous Iridium-Tellurium Alloy for Electrochemical N2 Reduction. J Am Chem Soc. 2023 Mar 22;145(11):6079-6086. doi: 10.1021/jacs.2c10637.

[67] Ma J, Shi T, Li Y, Yang B, Tian Y, Xu B, Yang H, Chen X, Chen C. Selective sulfidation-vacuum volatilization processes for tellurium and bismuth recovery from bismuth telluride waste thermoelectric material. J Environ Manage. 2023 Feb 1;327:116845. doi: 10.1016/j.jenvman.2022.116845.

[68] Liu D, Zha W, Yuan R, Lou B, Sa R. Indirect-to-direct band gap transition and optical properties of metal alloys of Cs2Te1-x Ti x I6: a theoretical study. RSC Adv. 2020 Oct 6;10(60):36734-36740. doi: 10.1039/d0ra07586h.

[69] Faizan M, Xie J, Murtaza G, Echeverría-Arrondo C, Alshahrani T, Bhamu KC, Laref A, Mora-Seró I, Haidar Khan S. A first-principles study of the stability, electronic structure, and optical properties of halide double perovskite Rb2Sn1-xTexI6 for solar cell applications. Phys Chem Chem Phys. 2021 Feb 28;23(8):4646-4657. doi: 10.1039/d0cp05827k.

[70] Yu D, Li Q, Zhang W, Huang S. Amorphous Tellurium-Embedded Hierarchical Porous Carbon Nanofibers as High-Rate and Long-Life Electrodes for Potassium-Ion Batteries. Small. 2022 Aug;18(32):e2202750. doi: 10.1002/smll.202202750.

[71] Zhang Y, Zhu H, Freschi DJ, Liu J. High-Performance Potassium-Tellurium Batteries Stabilized by Interface Engineering. Small. 2022 Apr;18(15):e2200085. doi: 10.1002/smll.202200085.

[72] Wu P, Mu Z, Qian K, Guo C, Li M, Li J. Biochar-Derived Hierarchical Porous Carbon as Tellurium Host for High-Performance Potassium-Tellurium Batteries. Chemistry. 2023 Dec 11;29(69):e202302121. doi: 10.1002/chem.202302121.

[73] Hellier K, Stewart DA, Read J, Sfadia R, Abbaszadeh S. Tuning Amorphous Selenium Composition with Tellurium to Improve Quantum Efficiency at Long Wavelengths and High Applied Fields. ACS Appl Electron Mater. 2023 May 3;5(5):2678-2685. doi: 10.1021/acsaelm.3c00150.

There is a lot of confusion also in chapters 6.1 and 6.2 with toxicity and biogeochemistry intermingled among the two.

Response: Thanks a lot for your comment. Improvements were made in these sections by re-arranging them to address the comment.

As you can understand, this is just a short list of suggestions that spring from several reading tiers of your second draft. Please, believe that I appreciate very much this nice compilation and highly esteem your efforts to make a point of more than 100 references on a niche topic. However, I still find the compilation, rather than the backgrounding science, still poorly organized. I would ask you to make the extra effort to rewrite a better draft, by re-organizing the contents in a smoother and better articulated way. It is a pity that this Journal does not allow Table of content in the reviews, because preparing that is very helpful in organizing the contents.

Response: Thanks a lot for your comment.

The table of contents for the MS that we work from is given below, and you are right, it is important to start the MS with the TOC, we totally agree with your opinion. We made changes and improvements in the TOC based on your comments as well.

  1. Introduction
  2. Discovery of tellurium
  3. Tellurium occurrence, forms in nature, and characterization

3.1 Tellurium occurrence

3.2 Global tellurium production

3.3 Tellurium forms in nature

3.4 Tellurium characterization

  1. Nanotellurium and its production
  2. Applications of tellurium and nanotellurium

5.1 Pharmaceutical applications

5.2 Biomedical applications

  1. Toxicity of tellurium and nanotellurium

6.1 Clinical signs

6.2 Biogeochemistry

6.3 Environmental effects

6.4 Toxicity of nano-tellurium

6.5 Tellurium toxicology and safety

  1. Conclusions

More improvements in many sections were added in the revised MS, hoping this will meet your approval, thanks

I again mark this draft for “major revision”, and I hope you understand the motivation: it is admirative, not despising! I strongly hope that you can make a much better, more informative review article to the benefit of many, many readers.

Response: Thanks a lot for your comments. We respect your opinion and your input will improve our MS.

Thanks again.

Kind regards.

Comments on the Quality of English Language

nothing more than check typos and go along with text revision.

Response: Thanks a lot for your comment. We checked again in the revised MS.

Round 3

Reviewer 2 Report

Comments and Suggestions for Authors

Dear Authors, dear Editor,

I think that this revision is all you can afford. I wouldn’t bother you further with improvements and suggest to have it published

Kind regards